Subject Areas:
physical chemistry/chemical physics

Keywords:
uranium-bearing sandstone, full-scale pore size distribution, nitrogen gas adsorption, nuclear magnetic resonance

Author for correspondence:
Sheng Zeng
e-mail: usczengs@126.com

# Full-scale pore size distribution features of uranium-bearing sandstone in the northwest of Xinjiang, China

Sheng Zeng[1], Hao Li[1], Ni Zhang[3], Bing Sun[2], Jinzhu Li[1] and Yulong Liu[4]

[1]School of Resources Environment and Safety Engineering, and [2]School of Civil Engineering, University of South China, Hengyang 421001, People's Republic of China
[3]Planning and Design Department, Chongqing GaoXin Engineering Survey and Design Institute Co., Ltd, Chongqing 401121, People's Republic of China
[4]Resource Business Department, CGNPC Uranium Resources Co., Ltd, Beijing 100029, People's Republic of China

SZ, 0000-0002-9283-6598

As an important nuclear fuel, uranium in sandstone uranium deposits is mainly extracted by *in situ* leaching. The porosity of sandstone is one of the important indexes determining *in situ* leaching efficiency. Moreover, the microscopic pore size distribution (PSD) of the uranium-bearing layer has an important effect on porosity. It is necessary to feature the pore structure by various techniques because of the different pore types and sizes in the uranium layer. In this paper, combined with nitrogen gas adsorption, nuclear magnetic resonance techniques and scanning electron microscopy, the full-scale PSD features of uranium-bearing sandstone in the northwest of Xinjiang are effectively characterized. The results show that pores structure of uranium-bearing sandstone include dissolution pores ($d \leq 50$ nm), intergranular pores (50 nm $< d \leq 200$ µm) and microfractures. Intergranular pores of 60 nm and 1 µm are the significant contributors to pore volume. The effects of the pore volume of two pore types (dissolution pores and intergranular pores) on the porosity of uranium-bearing sandstone are analysed. The results show that intergranular pores have the greater influence on the porosity and are positively correlated to the porosity. Dissolution pores have little effect on the porosity, but it is one of the key factors for improving uranium recovery. Moreover, the greater the difference of PSD between sandstones, the stronger the interlayer heterogeneity of uranium-bearing sandstone. This kind of interlayer

heterogeneity leads to the change of permeability in the horizontal direction of strata. It provides a basis for a reasonable setting of well type and well spacing parameters.

# 1. Introduction

Uranium is one of the main energy sources in the world [1,2]. Uranium minerals are rich in sandstone uranium deposits [3]. They have a pore-fracture double fractal structure, and their high complexity and anisotropy will greatly affect permeability [4,5]. In addition, during the *in situ* leaching of uranium, the leaching solution needs enough time and space to react chemically with sandstone uranium deposits [6]. Therefore, proper pore size distribution (PSD) and pore connectivity are required [7]. If there are a large number of large pores and pore throats in uranium-bearing sandstone, very good pore connectivity will be formed. It will cause the seepage rate of the leachate to be too fast ($k > 10.0 \, \mathrm{m \, d^{-1}}$), and thus cannot completely react with uranium minerals. On the contrary, if the seepage rate of leaching liquid is too slow ($k < 0.1 \, \mathrm{m \, d^{-1}}$), it is difficult to flow and migrate [8]. Thus, quantitative characterization of the distribution of the uranium-containing sand porosity is an important indicator of sandstone uranium mining method. At present, sandstone uranium ore in the mining process is rarely discussed at rock micro level. Few scholars have studied the complexity of ore pores through mercury intrusion test or scanning electron microscopy (SEM). At the same time, there are few studies on the pore division of uranium-bearing sandstones. Most scholars often use reservoir classification and relationship with uranium minerals to divide the sandstone porosity. Chen *et al.* [9] and Jiao *et al.* [10] divided the pore structure into primary pores, residual corrosion pores intergranular pores, secondary solution pores, intergranular pores and microcracks. Depending on the type of reservoir and the location of the sandstone area, there are different methods of classifying and describing pore structure. However, the pore type and pore size of the sandstone are not uniformly classified [11], which greatly hinders research with structural characteristics of uranium sandstone porosity.

Actually, many empirical approaches have been used for studying the pore structure of rocks in recent years. These pore research techniques are widely applied into the coal development, oil, natural gas and other resources. These methods include quantitative analysis and qualitative descriptions. Methods for quantitative analysis include mercury injection capillary pressure (MICP), nitrogen gas adsorption ($N_2GA$), small-angle neutron scattering (SANS) and nuclear magnetic resonance (NMR). They are used to calculate the pore specific surface area, pore volume and PSD. However, each method has advantages and limitations in characterizing the pore structure of sandstone [12]. For MICP, the pore space parameters can be directly obtained according to the rise and fall of mercury pressure, such as pores and pore throats number distribution, capillary pressure curve, etc. However, due to the shielding effect of small pore, the high-pressure intrusion of mercury may reduce the number of macropores, resulting in inaccurate PSD measurement [13,14]. The pore size obtained by $N_2GA$ is limited, ranging from 0.3 to 200 nm [15,16]. And $N_2GA$ needs vacuum drying treatment before testing, which may change pore structure [17]. Therefore, it is obliged to feature the micro-pore structure of the reservoir by combining $N_2GA$ and other means. SANS can provide connected pores and isolated pores of rocks, and is widely applied to pore structure research of coal seams, oil reservoirs and shale gas reservoirs [18–20]. However, it has high accuracy only in the nano scale pore size range, and cannot obtain a complete PSD [21]. Although NMR is a rapid and harmless method, the $T_2$ spectrum distribution curve requires to be translated into PSD. The minimum measurement aperture by NMR is determined by the echo distance and the relative coefficient of particle surface [22,23]. Methods for qualitatively describing the type, size and surface morphology of pores include SEM, micro-computed tomography (μ-CT) and other image analysis methods [24–27]. These methods are presented in the form of intuitive images and visualize the geometric shapes of pore and throat. It is beneficial to the construction of pore network model and the estimation of macroscopic permeability. However, because of the high cost of detection, they are seldom used in the testing of large quantities of specimens.

The pore size range of uranium-bearing sandstone is wide [28], and the PSD of rock cannot be accurately described by a single testing technology. Therefore, the full PSD of rocks can be revealed more effectively by the comprehensive utilization of various techniques [29–31]. Zheng *et al.* [32] used SEM, low-temperature nitrogen adsorption (LTNA) and NMR to characterize the full PSD of coal specimens. The SEM and LTNA can reveal the complex pore structure of coal and the NMR can

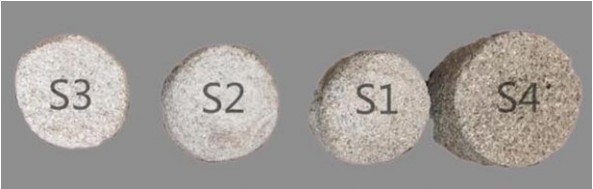

**Figure 1.** Experimental samples.

obtain PSD data, but the results were incomplete. Li *et al.* [33] pointed out that the complementarity of $N_2GA$ and NMR can be used to jointly reveal the full PSD in shale, but the method is incomplete.

This study takes sandstone samples from the sandstone-type uranium deposits in northwest Xinjiang as an example. Combined with $N_2GA$, NMR and SEM techniques, the complete PSD and pore type of uranium-bearing sandstone are determined. The pore structure features and anisotropism characteristics of this kind of sandstone are discussed. It will help to improve the systematic cognition of the pores of uraniferous sandstone. The study provides an experimental basis for the evaluation of uraniferous sandstone deposit, the effective exploration of uranium resources and the establishment of sandstone uranium deposit model.

# 2. Geological setting and property of sandstone specimens

Four specimens emanating from four neighbouring wells and different deepnesses in the Xinjiang uranium deposit were investigated in this study. Samples taken from the four boreholes are almost cylindrical and numbered S1, S2, S3 and S4, respectively, as shown in figure 1. Drilling depths are 458.5, 459.0, 455.5 and 459.8 m. The sample is cylindrical, 3.5–4.5 cm in radius and 4–5 cm in height. All specimens were professionally sealed and wrapped carefully with fresh-keeping film and foam mats to maintain their original shape. Afterwards, they were sent for fundamental petrophysical analysis. After 24 h of drying, the porosity of rocks was measured first, and then core plugs were divided into four parts and tested by XRD, SEM, NMR and $N_2GA$.

# 3. Pore structure experiments

In order to more accurately describe the pore structure characteristics of uranium-bearing sandstone. Four parallel control tests of XRD, SEM, NMR and $N_2GA$ were carried out on samples of uranium-bearing sandstone in Xinjiang, China. The pore size range obtained by various experimental methods is shown in figure 2. Four samples collected a small amount of powder to carry out XRD experiments to comprehend the mineral composition of the uranium-bearing sandstone specimens. It is helpful to classify and define the pore types in rocks.

## 3.1. X-ray diffraction

X-ray diffraction (XRD) experiments were conducted using a D8 ADVANCE diffractometer equipped with a Cu tube and a monochromator. Scan range was $2\theta$, 3–90° and scan interval was 0.02°. Samples are powdered and ground to about 300 meshes.

## 3.2. Scanning electron microscopy

SEM observations were conducted using FEI SCIOS, and the enlargement factors used in this research were between 200 and 10 000 times. The accelerating voltage and resolution were set to be 15 kV and 1.5 nm, respectively. The uranium-bearing sandstone specimens were anomalous and their edge length was about 1 cm. The polished samples were used for determining the pore structure of specimens by the environmental SEM.

## 3.3. Nuclear magnetic resonance

NMR measurements were conducted using MacroMR12–150H-I NMR core micro-nondestructive testing imaging and analysis system. The apparatus has a stationary magnetic field strength of 0.3 T, resonant

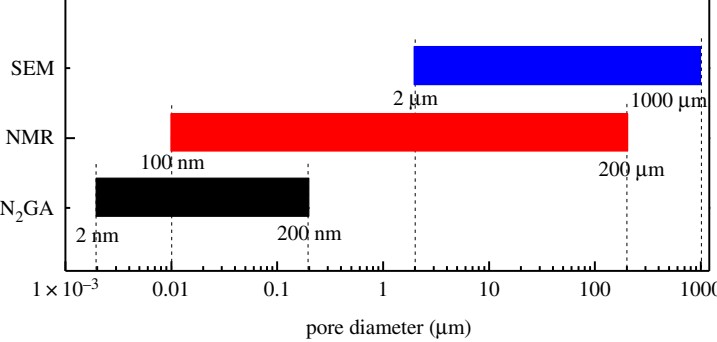

**Figure 2.** The pore size range obtained by various experimental methods.

frequency range of 1–42 MHz, and coil diameter is 60 mm. The samples were immersed in water for 24 h and then subjected to NMR. The samples were irregular and their volumes were 64.47, 45.47, 39.02 and 184.75 $cm^3$, respectively.

The spectral density of uranium-bearing sandstone cannot be obtained directly by NMR. According to the NMR relaxation theory, the relation between the radius of the pore and the $T_2$ value is as shown in equation (3.1) [34]

$$\frac{1}{T_2} \approx \rho_2 \left(\frac{S}{V}\right) = F_S \frac{\rho_2}{r}, \tag{3.1}$$

where $F_S$ is the pore geometry factor of uranium-bearing sandstone (spherical $F_S = 3$; columnar $F_S = 2$); it is assumed that the pore shape is spherical, that is, $F_S = 3$; $r$ is the radius of the pore of the uranium-bearing sandstone, μm; $S/V$ refers the pore specific surface area of the uranium-bearing sandstone, $cm^2\,g^{-1}$; $\rho_2$ denotes the transverse surface relaxation strength; $T_2$ denotes the transverse relaxation time, ms. On the basis of equation (3.1), the pore radius is proportional to the transverse relaxation time. According to the experimental $T_2$ spectrum data, the pore distribution of different pore sizes inside the sandstone can be qualitatively and quantitatively analysed.

## 3.4. Nitrogen gas adsorption

The $N_2$GA experiment was conducted by using a Tristar II 3020 Fully Automatic Surface and Pore Analyzer. The sandstone specimens were crushed into 60–80 mesh powder and then dried for 48 h. The dried powder was placed in a low-temperature environment for testing. The temperature of liquid nitrogen was controlled at 77 K and the relative pressure ($P/P_0$) between 0.001 and 1. The adsorption desorption isotherms of $N_2$ would be obtained. This technology is mainly used for testing nano scale pores within the tight sandstone. It can precisely and efficiently measure pores with radius of 2–200 nm. The specific experimental theory is based on the adsorption characteristics of the gas on the solid surface. Under a certain pressure, the particle surface of the tested sample has reversible physical adsorption on gas molecules at low temperature. There is some equilibrium adsorption capacity under the certain conditions. The equilibrium adsorption capacity was determined, and then the parameters were calculated by theoretical model [35]. Parameters assessed by this technique include: Brunauer–Emmett–Teller (BET) surface area, standard temperature and pressure (STP) quantity adsorbed, Barrett–Joyner–Halenda (BJH) adsorption/desorption average diameter, incremental pore volume, incremental pore area, etc. As for uranium-bearing sandstone with wide distribution of pore radius, other techniques are required to characterize the distribution of pore radius within rock samples.

# 4. Results and discussion

## 4.1. Pore types from SEM

Table 1 contains data on porosity and mineral composition of four uranium-bearing sandstone samples observed by SEM after making thin sections. The porosity was obtained by NMR and the mineral composition was obtained by XRD. Their porosity ranges from 16.94% to 20.09% with an average of 18.9% (±1.5%). It can be seen from table 1 that the uranium-bearing sandstone specimens were

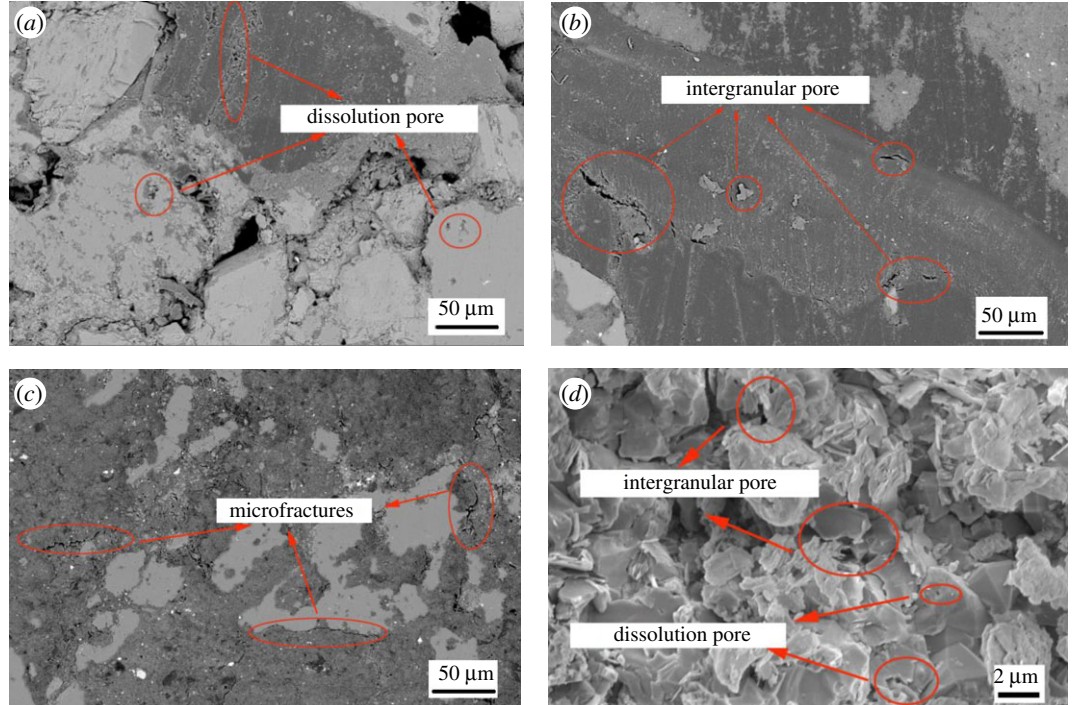

**Figure 3.** SEM images of selected uranium-bearing sandstone samples.

**Table 1.** Petrophysical parameters and mineral compositions of four tight sandstone samples.

| sample | depth (m) | porosity by NMR (%) | mineral composition by XRD (wt%) | | | | |
|---|---|---|---|---|---|---|---|
| | | | quartz | dolomite | calcite | clay minerals | others |
| S1 | 458.5 | 20.03 | 59.8 | 16.7 | 10.3 | 10.2 | 3.0 |
| S2 | 459.0 | 20.09 | 59.5 | 17.2 | 10.5 | 8.5 | 4.3 |
| S3 | 455.5 | 18.67 | 62.5 | 17.9 | 9.5 | 8.0 | 2.1 |
| S4 | 459.8 | 16.94 | 61.0 | 18.2 | 9.0 | 9.1 | 2.7 |

composed of quartz, dolomite, calcite and clay minerals. The main component is quartz with an average content of 60.7% (±1.4%).

The pores in the uranium-bearing sandstone are important circulation channels during the *in situ* leaching of uranium. According to the pore size and pore formation reasons, combined with SEM image recognition, the pore structure can be divided into three pore types, namely dissolution pores, pores between grains and microfissuring (figure 3). The size of intergranular pore is relatively large, which is the pore between particles. The pore diameter mainly ranges from 50 to 200 µm. The shapes of these holes are usually sharp and angular with smooth and straight polygons (figure 3*b,d*). Mechanical compaction, cementation and other late diagenesis resulted in intergranular pores, and are related to authigenic quartz (figure 3*b*. Dissolution pores (pore diameter less than or equal to 50 nm) are mainly anomalous intragranular pores (figure 3*a*), which has a smaller pore radius and poor connectivity. They are mainly the micro-pore formed by dissolving some minerals in the grain (figure 3*a,d*). The pore shape is similar to that of a circle and the edge is smooth (figure 3*a*). The more their number, the more favourable it is for leaching solution to replace uranium in sandstone.

Furthermore, there are a mass of microcracks (figure 3*c*) developed within the research area. There are many microcracks in this uranium sandstone. They are usually formed by extrusion and vibration of external forces such as geological tectonic movement, excavation and drilling. Its length is generally less than 200 µm, and its width is less than 5 µm. Affected by dissolution during diagenesis, the boundary of some microfractures widens to communicate different types of pores. On the one hand,

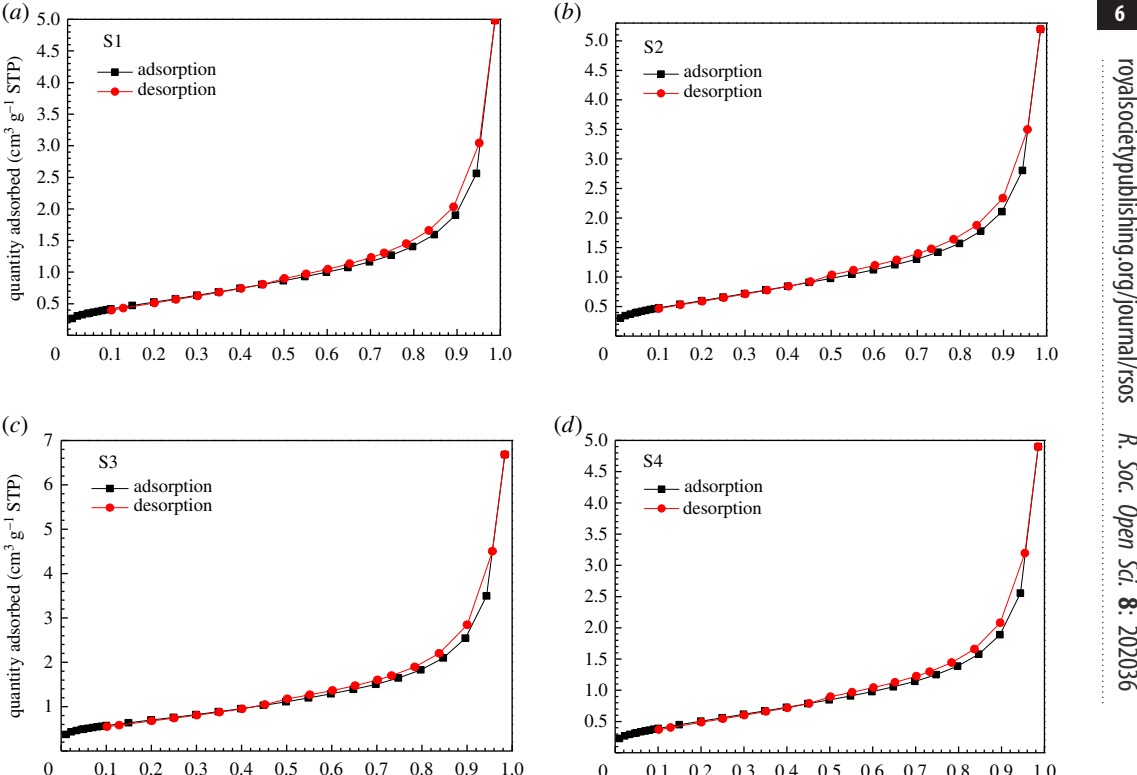

**Figure 4.** Adsorption–desorption isotherms for four uranium-bearing sandstone samples.

they can increase the permeability of sandstone and facilitate the migration of leaching solution within the rock. On the other hand, too many microfractures may cause cracks to penetrate, which will cause leakage in the leaching process and pollute groundwater and soil environment.

## 4.2. Pore size distributions

### 4.2.1. Analysis of $N_2$ adsorption/desorption isotherms

$N_2$ adsorption/desorption isotherms of four uranium-bearing sandstone specimens under specific conditions are shown in figure 4. All nitrogen adsorption isotherms are type IV under the criteria of International Union of Pure and Applied Chemistry (IUPAC) [36], and their pore diameters are in the range of dissolution pores ($2 < d \leq 50$ nm). If the relative pressure is less than 0.1, the adsorption curve is gentle and slightly raised upward. If the relative pressure is between 0.1 and 0.7, the adsorption curve is almost linear, the amount of nitrogen adsorption is small. If the relative pressure approaches 1.0, the nitrogen adsorption amount steadily rises. It adsorbs on the macrovoids, and nitrogen adsorption rapidly increases with the increasing of pressure. The curve rises abruptly and the horizontal platform does not appear, indicating that the uranium-containing sandstone sample has not reached the maximum adsorption capacity. This phenomenon indicates that the sandstone samples studied also contained a range of intergranular pores ($d > 50$ nm) [37], which were not detectable by $N_2$GA experiments.

Due to the capillary condensation occurring in the adsorption, the adsorption isotherm increased sharply, and hysteresis was observed in the higher $P/P_0$ region. That is, the adsorption isotherm is lower than the desorption isotherm, and a desorption hysteresis occurs, showing a hysteresis loop. Further analysis of the hysteresis loop morphology of the nitrogen adsorption/desorption curve can comprehend the pore morphological festures of sandstones deeper. Combined with the five types of hysteresis rings classified by IUPAC and the adsorption/desorption isotherm curve characteristics (figure 4), it can be determined that the hysteresis rings of S1, S2, S3 and S4 are all the third type. This shows that such uranium-bearing sandstone have dissolution pores with smaller pore sizes and mainly contain parallel plate-shaped pores with good development potential.

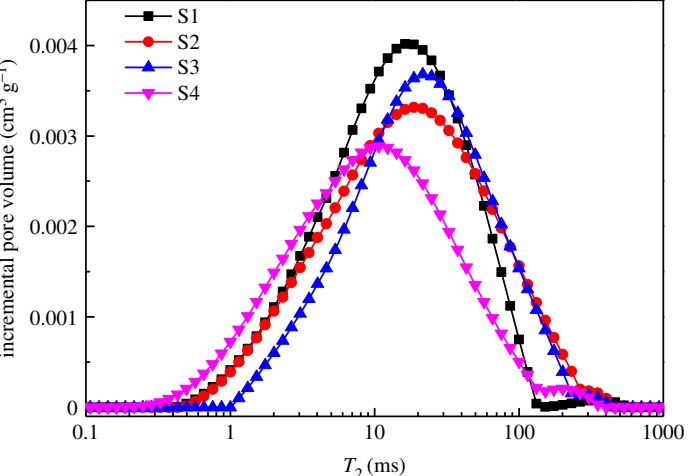

**Figure 5.** $T_2$ spectrum curves of water saturated for four uranium-bearing sandstone samples.

### 4.2.2. Analysis of NMR $T_2$ distributions

The pore shape of uranium-bearing sandstones is deemed to be columniform, which keeps abreast of the pore model of Washburn equation [34]

$$P_c = \frac{2\sigma\cos\theta}{r}, \tag{4.1}$$

where $P_c$ is the capillary pressure, $\theta$ is the contact angle, $r$ is the tube radius and $\sigma$ is the surface tension of the wetting liquid.

According to equations (3.1) and (4.1), $F_s$ is equivalent to 2. As shown in figure 5, the $T_2$ spectra of four uranium-bearing sandstone samples were obtained by NMR experiments. All $T_2$ spectra have two peaks with different amplitudes on the left and right sides, and the right peak is obviously smaller than the left peak. $T_2$ distributions of all specimens primarily range 0.1 to 1000 ms and move to a smaller value from specimen S1 to S4 (figure 5). The change of $T_2$ NMR spectrum can reflect the change of sandstone microstructure. It also can qualitatively and quantitatively describe the change of pore size and connectivity in the sandstone.

Equation (3.1) shows that the pore size of sandstone is proportional to relaxation time $T_2$. The smaller $T_2$ is, the smaller pore diameter is. The pore radius corresponding to the peak position in $T_2$ spectrum is directly related to the pore type. The left peak region represents intergranular pores and dissolution pores (0.02 μm < $d$ < 5 μm), and the right peak region represents intergranular pores (5 μm < $d$ < 11 μm). The left peak value of $T_2$ spectrum is obviously higher than the right peak, reflecting that the pore diameter of rock is primarily concentrated in about 1 μm. The peak area of the left peak of $T_2$ spectrum is obviously greater than half of the total area. It reflects that the intergranular pore is the main pore type existing in the uranium-bearing sandstone. The wider $T_2$ distribution indicates that the samples have better connectivity [38]. The range of pore diameters obtained by NMR is wide, but the characterization of small pores is inaccurate. Therefore, the time of echo (TE) needs to be small enough to find smaller pores [39].

### 4.2.3. Determination of the full-scale PSD by combining N$_2$GA and NMR techniques

Figure 6 shows the full-scale PSD obtained from N$_2$GA and NMR techniques. The PSD of all samples is bimodal. Pore diameter of the smaller peaks is about 4 nm, the larger peak is about 60 nm, and the main pore size range is 3–200 nm. However, the pore volume corresponding to the smaller peak is obviously very small, so this analysis of PSD can ignore it (figure 6). The PSDs measured by NMR technology also show a bimodal pattern. The pore diameter of the smaller peak is about 1 μm, and that of the larger peak is about 10 μm. The main pore size range is 0.02–20 μm.

In summary, the complete pore size of uranium-bearing sandstone is widely distributed, and the pore size ranges from 2 nm to dozens of micrometres. It is difficult to accurately describe the pore characteristics of uranium-bearing sandstone without any other test method. $T_2$ distribution is obtained with NMR test, which reflects the PSD and obtains other physical parameters of the sandstone [40]. However, the accuracy in the mesoporous and microporous regions is relatively low, and the nitrogen adsorption test can make up

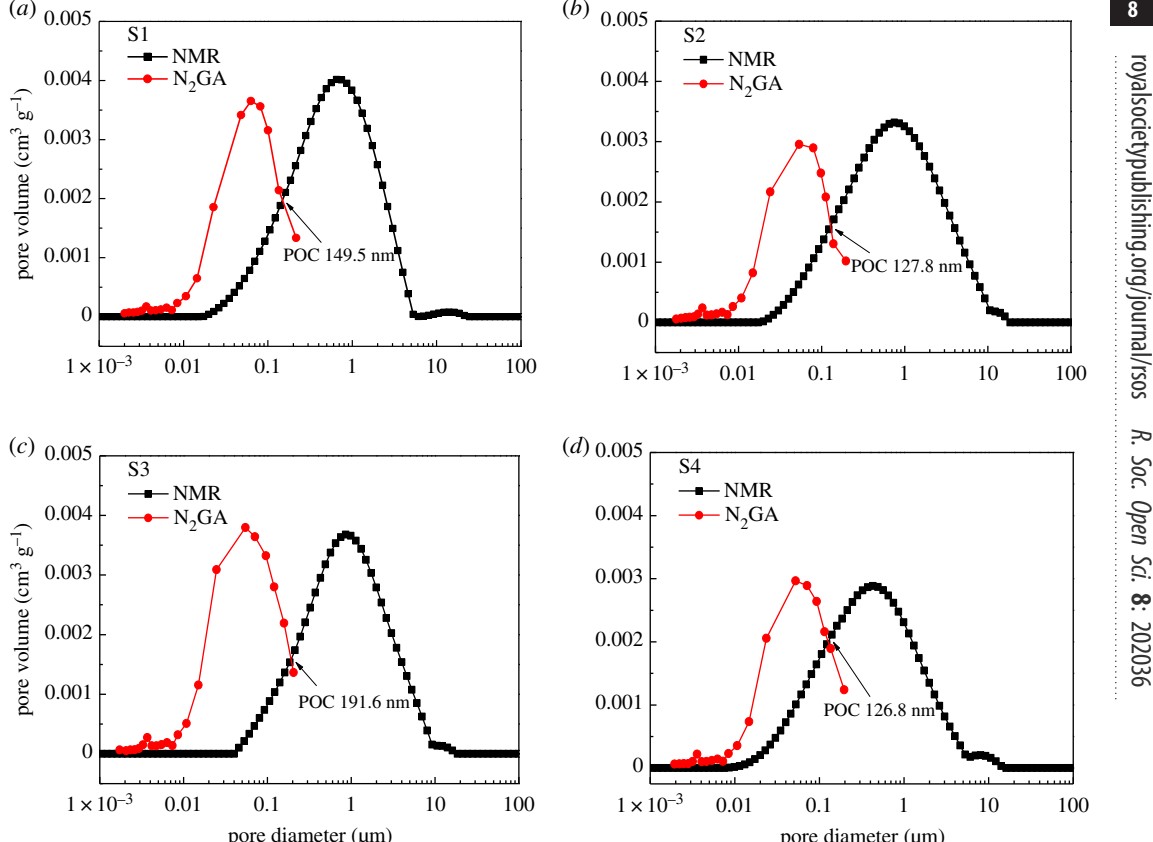

**Figure 6.** Pore volume curves obtained from N$_2$GA and NMR techniques.

for this defect. Therefore, the combination of N$_2$GA and NMR techniques can be considered as effective methods to determine the total PSD of uranium-bearing sandstone. In N$_2$GA test, the equivalent pore size ranges from 2 to 200 nm. NMR tests pore size between tens of nanometres and tens of micrometres, and the more the pore size reaches the critical range, the lower the measurement accuracy. Therefore, there is an overlapping range of N$_2$GA and NMR, and there is a need to confirm the point of attachment (POC) of the two technologies. The method argues that the pore volume measured by different means should be equal under the same pore diameter, that is, $dV_{N2GA} = dV_{NMR}$. Therefore, POC is the intersection of two curvilinears from N$_2$GA and NMR laboratory data (figure 6). However, this method only considers the relevance of the volume increment and the corresponding bore diameter. It ignored the respective characteristics of N$_2$GA and NMR, so there are still some errors [41].

From figure 5, it can be seen that both N$_2$GA and NMR are open-downward curves, and intersect with the rising stage of NMR in the decreasing stage of N2GA PSD curve. This is because the measurement range of the two techniques coincides at 20–200 nm, and the experimental accuracy of N$_2$GA is higher in this range. With the increase of hole diameter (greater than 200 nm), the experimental accuracy of NMR is gradually improved. Therefore, by combining the N$_2$GA data on the left side of POC with the NMR data on the right side, the complete PSD can be determined by the following steps. Firstly, the two types of experimental data are converted into the relation of the pore diameter and the corresponding pore volume. Then the intersection point POC of the two curves is found, and the intersection pore diameter and the corresponding pore volume are obtained. Finally, the full PSD is obtained by removing the overlapping data.

Figure 7 shows full PSDs of the four uranium-bearing sandstone specimens determined by this method, and table 2 shows the porosity obtained by cumulative calculation of full PSD curves. The porosity obtained by combination method is obviously larger than either method alone. It is closer to that obtained by NMR. It indicates that combined N$_2$GA and NMR technology can more accurately feature the full PSD of uraniferous sandstone. Figure 7 also shows that the pore size of the sandstone in this type of uranium deposit mainly concentrates at about 60 nm and 1 µm. Its distribution range is wide, ranging from 3 nm to 20 µm, which has a significant impact on the formation of porosity. With the increasing of porosity, the bore diameter corresponding to the double peaks of the full PSDs curve

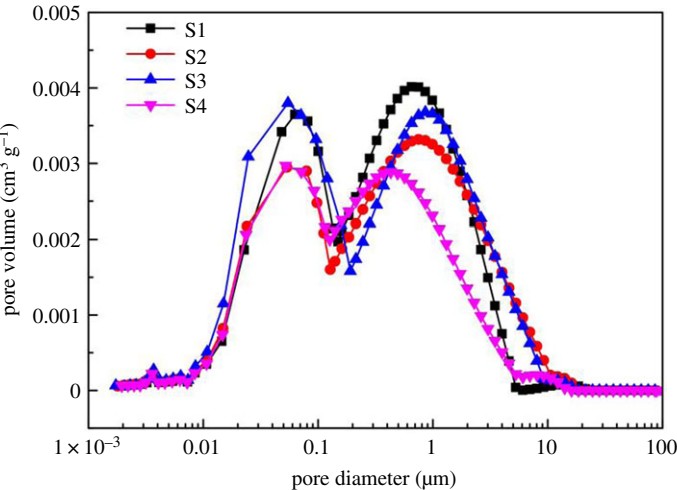

**Figure 7.** The full pore size distributions of uranium-bearing sandstone.

**Table 2.** Porosity of uranium-containing sandstone measured by $N_2GA$ and NMR.

| sample | porosity of $N_2GA$ (%) | porosity of NMR (%) | combination of $N_2GA$ and NMR | |
|---|---|---|---|---|
| | | | porosity (%) | POC (nm) |
| S1 | 4.95 | 20.03 | 22.60 | 149.5 |
| S2 | 5.20 | 20.09 | 22.08 | 127.8 |
| S3 | 6.71 | 18.67 | 22.62 | 191.6 |
| S4 | 4.86 | 16.94 | 17.68 | 126.8 |

increases by degrees, and the peak value also increases significantly. In other words, with increasing porosity, the larger the pore size, the higher the permeability [42,43]. The larger the bore diameter, the more obvious the effect of the pore on the fluid migration in the uranium-bearing sandstone.

From the pore type, the diameter corresponding to the two peaks (about 60 nm and about 1 µm) is within the size range of intergranular pores (50 nm < pore diameter ≤ 200 µm). It indicates that the uranium-bearing sandstone is mainly consisting of intergranular pores. The content of dissolution pores is significantly lower than intergranular pores. The pore volume of these two types of intergranular pores is similar, and the disparity of pore size is large. There is enough time and space for uranium replacement reaction, and it is conducive to the migration of uranium-bearing leach solution in uranium-bearing sandstone. Therefore, this type of uranium-bearing sandstone meets the conditions of *in situ* leaching and has high uranium recovery.

Because the permeability of uranium-bearing sandstone is controlled by its PSD position, pore network model can be constructed by using full PSDs data. The permeability and seepage velocity of uranium-bearing sandstone can be predicted and an optimal full PSDs range can be found for *in situ* leaching. Therefore, through the sandstone mining of the same type of uranium deposit, the best distribution range is compared, and the pore distribution of uranium-bearing sandstone is obtained. The targeted measures can be formulated in advance to obtain the best mining scheme. For example, if there are more dissolution pores and smaller porosity, the number of intergranular pores can be increased by increasing the injection pressure or pumping speed. On the contrary, when there are more intergranular pores, the time of uranium leaching can be increased by slowing down the pumping speed or expanding the well spacing.

## 4.3. Analysis of pore structure characteristics

### 4.3.1. Analysis of pore structure characteristic parameters

Cumulative pore volume, total pore volume, average pore diameter and maximum pore radius of different pore types can reflect the pore characteristics of the uranium-bearing sandstone ore layers. It

**Table 3.** Pore structure parameters of uranium-bearing sandstone samples.

| sample | cumulative pore volume of dissolution pore ($cm^3 g^{-1}$) ($d \leq 50$ nm) | cumulative pore volume of intergranular pore ($cm^3 g^{-1}$) (50 nm $< d \leq$ 200 μm) | total pore volume ($cm^3 g^{-1}$) | average pore radius (μm) | maximum pore throat radius (μm) | average pore throat radius (μm) |
|---|---|---|---|---|---|---|
| S1 | 0.0112 | 0.0820 | 0.0932 | 0.21 | 13.30 | 0.58 |
| S2 | 0.0080 | 0.0803 | 0.0883 | 0.28 | 9.15 | 0.82 |
| S3 | 0.0104 | 0.0797 | 0.0901 | 0.29 | 7.52 | 0.80 |
| S4 | 0.0075 | 0.0632 | 0.0707 | 0.17 | 6.83 | 0.57 |

also can predict the seepage velocity and uranium mining efficiency of the sandstone. The pore structure characteristic parameters are detailed in table 3. The cumulative pore volume can be obtained by integrating the distribution curve of pore diameter (figure 7). The sum of the cumulative pore volumes of the two types are equal to the real pore volume. The maximum and average pore throat radius are obtained by NMR.

According to parameters of pore structure (table 3), the cumulative pore volume of the dissolution pores ranges from 0.0075 to 0.0112 $cm^3 g^{-1}$, average is 0.0093 (±0.0018) $cm^3 g^{-1}$. The cumulative pore volume of intergranular pores ranges from 0.0632 to 0.0820 $cm^3 g^{-1}$, average is 0.0763 (±0.0088) $cm^3 g^{-1}$. The total pore volume of uranium-bearing sandstone ranges from 0.0707 to 0.0932 $cm^3 g^{-1}$, and the average pore volume is 0.0856 (±0.0101) $cm^3 g^{-1}$. The average pore radius is between 0.17 and 0.29 μm, average is 0.24 (±0.06) μm. Maximum pore throat radius is 6.83–13.30 μm, average is 9.2 (±2.9) μm. The average pore throat radius is 0.57–0.82 μm, and the average is 0.69 (±0.14) μm. The pore characteristic structure parameters shown in table 3 are all calculated under the full-scale PSD. The structure parameters can be used for studying the correlation between pore parameters and increasing the accuracy of pore structure characteristic analysis.

Permeability is the critical element to determine whether *in situ* leaching is adopted in uranium deposits, because it is related to the leaching efficiency of uranium elements. Moreover, rock permeability is generally related closely to porosity. The larger the porosity, the better the permeability, and the smaller the porosity, the worse the permeability. It has been proven in many related fields, such as coal rock, shale, gas sandstone, etc. [44–46]. Therefore, the relationship between porosity and accumulative volume of dissolution pore and intergranular pore can be explored to better estimate permeability of uranium-bearing sandstone. The porosity is a positive relationship with the cumulative pore volume of intergranular pore, but not with the dissolution pore. It shows that intergranular pore is an important index to determine the size of porosity, and acts as the main flow channel, which is consistent with the conclusion in reference [21].

According to the formation mechanism of dissolution pore, it is known that dissolution pore mainly exists in clay minerals and enriches a great deal of uranium. And the dissolution pores increase the specific surface area of the sandstone pores and the contact area between the leaching solution and the sandstone. It is conducive to obtaining more uranium. It shows that dissolution holes are one of the key factors affecting the recovery of sandstone-type uranium deposits. The pores are the relatively bulky parts enclosed by the framework grains, which has an important effect on fluid storage. Their throat is a relatively narrow part with a small pore volume. It has important impact on communicating the pores. Therefore, the permeability of sandstone layer is controlled mainly by throat size. The maximum pore throat radius obtained by NMR detection is in the range of 6.8–13.3 μm, average is ratio of 0.215, and the average pore throat diameter is between 1.2 and 1.8 μm. The result indicates that there are many large pore throats in this uranium deposit, which is conducive to the migration and accumulation of fluids and has good permeability.

From the full PSD curve in figure 7 and table 3, the average pore radius in this uranium deposit is between 0.17 and 0.29 μm. It is more accurate than the single test result. And pores in this pore size range are favourable for uranium leaching and uranium migration in uranium-bearing sandstone. It is worth mentioning that the number of samples in these experiments is limited. This only represents the change trend of porosity with cumulative pore volume, and the quantitative correlation needs further study.

### 4.3.2. Analysis of heterogeneous parameter

The heterogeneity of micro-nano scale pores in uranium-bearing sandstone greatly affects the permeability of uranium deposits [47]. For this article, the parameters of pore structure obtained by $N_2GA$ and NMR experiments were treated by using the method of studying the microscopic heterogeneity of unconventional oil and gas reservoirs. From the three heterogeneous parameters of relative deviation, coefficient of variation and range, the anisotropism of the pore structure is described as average pore size, pore volume, cumulative volume of dissolution hole and cumulative volume of intergranular hole.

#### 4.3.2.1. Relative deviation

Relative deviation refers to the percentage of the absolute deviation of a test in the mean value of multiple tests, denoted as $R_d$. The relative deviation can be used to measure the deviation degree of a certain parameter from its mean value, expressed as equation (4.2) [48],

$$R_d = \frac{|x_i - x_{\text{avg}}|}{x_{\text{avg}}}, \tag{4.2}$$

where $x_i$ represents the pore structure parameter of uranium-bearing sandstone, and $x_{\text{avg}}$ is the average value of a parameter representing the pore structure.

#### 4.3.2.2. Variable coefficient

Coefficient of variation is the ratio of the standard deviation of a parameter in the uraniferous sandstone's pore structure to its mean value, denoted as $C_v$. The variation coefficient can eliminate the influence of measurement scale and dimension, so as to compare the dispersion degree of two groups of parameters, expressed as equation (4.3) [49],

$$C_v = \frac{\sqrt{1/n \sum_{i=1}^{n} (x_i - x_{\text{avg}})^2}}{x_{\text{avg}}}. \tag{4.3}$$

#### 4.3.2.3. Range

Range is equal to the maximum minus minimum of a parameter, denoted by $R$. The extreme difference indicates the dispersion range and the pore structure parameters' difference degree. because of the different parameter units, the value difference is large, so the extreme difference is calculated by dividing the maximum value by the minimum. The dispersion degree is smaller when the ratio is closed to 1, and the better the homogeneity of the pore structure of uranium-bearing sandstone is. $R$ is expressed as equation (4.4),

$$R = \frac{x_{\text{max}}}{x_{\text{min}}}, \tag{4.4}$$

where $x_{\text{max}}$ and $x_{\text{min}}$ are, respectively, the maximum and minimum values of a certain type of parameter that characterizes the pore structure characteristics of uranium-bearing sandstone.

According to equations (4.2)–(4.4), substituting the pore structure characteristic parameter data gives the relative deviation, coefficient of variation and range difference. The specific values are shown in tables 4 and 5.

According to table 4, the variation trend of relative deviation of different pore structure parameters is not consistent. For example, the sample with the smallest relative deviation of the total pore volume may not have the smallest relative deviation of the average pore diameter. From the horizontal view of table 4, the relative deviation values of the same kind of parameters are all small, but the fluctuation range is large. The results indicate that the pore morphology of this kind of uraniferous sandstone is different and changeable, and the pore structure is complicated.

The heterogeneity of microscopic pore structure refers to the influencing factors that affect the micro-pore structure, distribution and connectivity. The coefficient of variation and range reflect the fluctuation degree and range of the pore structure parameters, it also reflects the degree of pore development, and then the heterogeneity of microscopic pores and the seepage characteristics are obtained. As shown in table 5, the variation coefficient and range of the cumulative pore volume of the dissociated pores are larger than that of intergranular pores. It shows that the difference degree

**Table 4.** The relative deviation of pore structure characteristic parameters.

| pore structure characteristic parameters | S1 | S2 | S3 | S4 |
|---|---|---|---|---|
| average pore size | 0.13 | 0.17 | 0.21 | 0.29 |
| total pore volume | 0.09 | 0.03 | 0.05 | 0.17 |
| cumulative pore volume of dissolution pores | 0.0019 | 0.0013 | 0.0011 | 0.0018 |
| cumulative pore volume of intergranular pores | 0.0057 | 0.0040 | 0.0034 | 0.0131 |

**Table 5.** The variable coefficient and range of pore structure characteristic parameters.

| pore structure characteristic parameters | variable coefficient | range |
|---|---|---|
| average pore size | 0.21 | 0.706 |
| total pore volume | 0.10 | 1.318 |
| cumulative pore volume of dissolution pores | 0.17 | 1.493 |
| cumulative pore volume of intergranular pores | 0.10 | 1.297 |

of dissolution pores is larger than that of intergranular pores, the dispersion range is wider, and the development degree is more uneven. The analysis shows that the smaller the bore diameter is, the greater the fluctuation of its pore structure parameters will be; that the stronger the micro-pore structure heterogeneity, the greater the resistance to the leaching solution flow, and the higher the uranium leaching concentration.

The variation coefficient and range of the total pore volume are located between that of the corrosion holes and the intergranular pores. This shows that the uneven development of dissolution pores is neutralized by intergranular pores, which results in the total pore volume of diverse samples getting closer. The coefficients of variation and range of the total pore volume and mean pore size are small. It indicates that the homogeneity of each sample is not much different. That is, the interlayer homogeneity of the uranium deposit is good, and the flow velocity in each seepage direction is not much different. The same well pattern and spacing can be used. Therefore, before the uranium deposit is mined, the microporous heterogeneity and interlayer heterogeneity can be obtained through the analysis of heterogeneity parameters. By judging the changes in porosity and penetration, the direction of percolation can be predicted and the well spacing can be reasonably arranged. This will increase the spread of the leaching solution and improve the uranium mining effect.

# 5. Conclusion

The combination of SEM, $N_2$GA and NMR is of great significance to reveal the pore structure features of uranium-bearing sandstone in northwest Xinjiang. This paper draws the following conclusions:

(1) Uranium-bearing sandstones include intergranular pores, corrosion holes and microfractures observed by SEM images. The pore size of the dissolution pore is the smallest, which is mainly concentrated within 50 nm. It is formed by chemical erosion and has poor connectivity. The pore size of intergranular pore ranges from 50 nm to 200 µm, which is formed by diagenesis and has good connectivity. Micro fractures are irregular linear, generally less than 200 µm in length and less than 5 µm in width. They are formed by external force and have the best connectivity.

(2) This study further proves that the combination of $N_2$GA and NMR is a reliable method to characterize the full-scale PSD of uranium-bearing sandstone. The studied uranium-bearing sandstones have a wide pore distribution characteristic between 3 µm and tens of micrometres. The cumulative pore volume of intergranular pores of about 60 nm and 1 µm is the largest, which mainly affects the porosity of sandstone. Moreover, the cumulative pore volume of intergranular pores are positively correlated with porosity. It is the main contributor of permeability just like microfractures. There is no correlation between dissolution pore and porosity. However, it

increases the contact area between solution and sandstone, which has important influence on recovery of sandstone-type uranium deposits.

(3) The intergranular pores and dissolution pores have a balanced development degree and good homogeneity. It is conducive to the seepage of the leaching solution. Moreover, the smaller the pore size of sandstone, the stronger the heterogeneity of micro pores and the worse permeability. The larger the difference of pores size distribution among sandstones, the stronger the interlayer heterogeneity of uranium-bearing sandstone. The interlayer homogeneity of this kind of uranium-bearing sandstone is good. The difference in porosity and permeability is not large. The seepage trace is relatively uniform. The same well spacing can be used to improve the production efficiency.

Data accessibility. The datasets supporting this article have been uploading as the electronic supplementary material, and they can also be found on https://doi.org/10.5061/dryad.c59zw3r61. The data are provided in electronic supplementary material [50].

Authors' contributions. S.Z. and N.Z. carried out the laboratory work, conceived of the study and designed the study; N.Z. and H.L. participated in data analysis, participated in the design of the study and drafted the manuscript; J.L. coordinated the study and helped draft the manuscript; B.S. revised it critically for important intellectual content; Y.L. provided experimental samples and related information to assist in the preliminary work of the experiment. All authors gave final approval for publication.

Competing interests. The authors declare no competing interests.

Funding. Financial support came from the Scientific Research Fund of the National Natural Science Foundation of China (grant no. 11775107), and the Key Projects of Education Department of Hunan Province of China (grant no. 16A184).

Acknowledgements. The authors would like to acknowledge the support of the Scientific Research Fund of the National Natural Science Foundation of China (no. 11775107), and the Key Projects of Education Department of Hunan Province of China (no. 16A184).

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
