## [Peer Review File · Royal Society Open Science]

Review History

RSOS-202036.R0 (Original submission)

Review form: Reviewer 1

Is the manuscript scientifically sound in its present form?

No

Are the interpretations and conclusions justified by the results?

Yes

Is the language acceptable?

No

Do you have any ethical concerns with this paper?

No

Have you any concerns about statistical analyses in this paper?

No

Recommendation?

Major revision is needed (please make suggestions in comments)

Comments to the Author(s)

Dear Authors,

You can find my suggestions and comments below:

Please get help from someone with full professional proficiency in English.

There are long sentences in the manuscript which makes it hard to follow. Please use shorter sentences.

Line 64: Please explain what PSD measurement is.

Line 103: "SEM, N2GA and NMR": Please change the order based on the appearance in the manuscript. It should be as "XRD, SEM, N2GA and NMR".

Under 2.21: Please use two paragraphs.: One for XRD explanations and another one for SEM explanations.

Line 132: What is the pressure unit?

Line 138: BET, STP, BJH: Please use the full names and give the abbreviations in parenthesis.

Lines 81, 119, 132, 137 140, 190, 215, 229, 252: "We": Please do not use active voice in the manuscript, you should use passive voice.

Table 3: Please do not use Chinese in the manuscript, change it to English.

Line 189: Explain the first sentence: What is Washburn equation?

Line 203: What does very good mean quantitatively for peaks of T2 spectrum and connectivity?

Figure 7: Does magenta curve belong to Sample 4? It has been written S7. Please fix this issue.

While giving the average values in the text, include standard deviation in the parentheses (i.e., Average (\pm SD) Unit)

Please fix, misspelled words in the text.

Line 373 Please explain your last conclusion paragraph sentences. "The interlayer homogeneity of this kind of uranium-bearing sandstone is good, and the difference in porosity and permeability is not large. The same well spacing can be used to improve the production efficiency."

Review form: Reviewer 2 (Lai Jin)

Is the manuscript scientifically sound in its present form?

Yes

Are the interpretations and conclusions justified by the results?

Yes

Is the language acceptable?

Yes

Do you have any ethical concerns with this paper?

No

Have you any concerns about statistical analyses in this paper?

Yes

Recommendation?

Accept with minor revision (please list in comments)

Comments to the Author(s)

Below are my minor comments

1. The first paragraph of Introduction, too long, cut down

2. paragraph 2, about the methods in pore structure characterization, see: Lai J., Wang G., Wang Z., Chen J., Pang X., Wang S., Zhou Z., He Z., Qin Z., Fan X. 2018. A review on pore structure characterization in tight sandstones. *Earth-Science Reviews*, 177, 436–457.
3. Section 2.1 Geological settings should be a new section
4. Section 3. Had better divided results from Discussion
5. Section 3.1 I think you firstly should use thin sections
6. There are many long paragraphs
7. NMR capture continuous whole pore body distribution (see: Lai J., Wang S., Zhang C., Wang G., Song Q., Chen X., Yang K., Yuan C. 2020. Spectrum of pore types and networks in the deep Cambrian to Lower Ordovician dolostones in Tarim Basin, China. *Marine and Petroleum Geology*, 112, 104081.), but N₂g is not, how to combine
8. too many equations in the paper, and references may need if they are not developed by the authors themselves

Decision letter (RSOS-202036.R0)

Dear Dr Zeng

The Editors assigned to your paper RSOS-202036 "Full-scale pore size distribution features of Uranium-bearing sandstone in the northwest of Xinjiang, China" have now received comments from reviewers and would like you to revise the paper in accordance with the reviewer comments and any comments from the Editors. Please note this decision does not guarantee eventual acceptance.

Please submit your revised manuscript and required files (see below) no later than 21 days from today's (ie 25-Mar-2021) date. Note: the ScholarOne system will 'lock' if submission of the revision is attempted 21 or more days after the deadline. If you do not think you will be able to meet this deadline please contact the editorial office immediately.

on behalf of Professor Zach Agioutantis (Associate Editor) and R. Kerry Rowe (Subject Editor)
 openscience@royalsociety.org

Reviewer comments to Author:

Reviewer: 1

Comments to the Author(s)

Dear Authors,

You can find my suggestions and comments below:

Please get help from someone with full professional proficiency in English.

There are long sentences in the manuscript which makes it hard to follow. Please use shorter sentences.

Line 64: Please explain what PSD measurement is.

Line 103: "SEM, N2GA and NMR": Please change the order based on the appearance in the manuscript. It should be as "XRD, SEM, N2GA and NMR".

Under 2.21: Please use two paragraphs.: One for XRD explanations and another one for SEM explanations.

Line 132: What is the pressure unit?

Line 138: BET, STP, BJH: Please use the full names and give the abbreviations in parenthesis.

Lines 81, 119, 132, 137 140, 190, 215, 229, 252: "We": Please do not use active voice in the manuscript, you should use passive voice.

Table 3: Please do not use Chinese in the manuscript, change it to English.

Line 189: Explain the first sentence: What is Washburn equation?

Line 203: What does very good mean quantitatively for peaks of T2 spectrum and connectivity?

Figure 7: Does magenta curve belong to Sample 4? It has been written S7. Please fix this issue.

While giving the average values in the text, include standard deviation in the parentheses (i.e., Average (\pm SD) Unit)

Please fix, misspelled words in the text.

Line 373 Please explain your last conclusion paragraph sentences. "The interlayer homogeneity of this kind of uranium-bearing sandstone is good, and the difference in porosity and permeability is not large. The same well spacing can be used to improve the production efficiency."

Reviewer: 2

Comments to the Author(s)

Below are my minor comments

1. The first paragraph of Introduction, too long, cut down
2. paragraph 2, about the methods in pore structure characterization, see: Lai J., Wang G., Wang Z., Chen J., Pang X., Wang S., Zhou Z., He Z., Qin Z., Fan X. 2018. A review on pore structure characterization in tight sandstones. *Earth-Science Reviews*, 177, 436-457.
3. Section 2.1 Geological settings should be a new section
4. Section 3. Had better divided results from Discussion
5. Section 3.1 I think you firstly should use thin sections

6. There are many long paragraphs
7. NMR capture continuous whole pore body distribution (see: Lai J., Wang S., Zhang C., Wang G., Song Q., Chen X., Yang K., Yuan C. 2020. Spectrum of pore types and networks in the deep Cambrian to Lower Ordovician dolostones in Tarim Basin, China. *Marine and Petroleum Geology*, 112, 104081.), but N2ga is not, how to combine
8. too many equations in the paper, and references may need if they are not developed by the authors themselves

===PREPARING YOUR MANUSCRIPT===

===PREPARING YOUR REVISION IN SCHOLARONE===

Author's Response to Decision Letter for (RSOS-202036.R0)

See Appendix A.

Decision letter (RSOS-202036.R1)

Dear Dr Zeng,

It is a pleasure to accept your manuscript entitled "Full-scale pore size distribution features of Uranium-bearing sandstone in the northwest of Xinjiang, China" in its current form for publication in Royal Society Open Science.

Please ensure that you send to the editorial office an editable version of individual files for each figure and table included in your manuscript. You can send these in a zip folder if more convenient. Failure to provide these files may delay the processing of your proof.

You can expect to receive a proof of your article in the near future. Please contact the editorial office (openscience@royalsociety.org) and the production office (openscience_proofs@royalsociety.org) to let us know if you are likely to be away from e-mail contact – if you are going to be away, please nominate a co-author (if available) to manage the proofing process, and ensure they are copied into your email to the journal.

on behalf of Professor Zach Agioutantis (Associate Editor) and R. Kerry Rowe (Subject Editor)
openscience@royalsociety.org

Appendix A

Detail Response to Reviewers

Reviewer: 1

Comments:

Q1: Please get help from someone with full professional proficiency in English.

Reply:

We appreciate you very much for the valuable comments. This manuscript has been helped by professionals who are proficient in English.

Q2: There are long sentences in the manuscript which makes it hard to follow. Please use shorter sentences.

Reply:

Thanks for your valuable comments. The long sentences in the manuscript have been replaced with short sentences. The modified sentences are readable(Please see Lines #19-21, 22-24, 48-50, 51-53, 64-66, etc.).

Q3:Line 64: Please explain what PSD measurement is.

Reply:

Thanks for your valuable suggestions. PSD measurement refers to pore size distribution measurement (Please see Lines #54-55).

Q4: Line 103: “SEM, N2GA and NMR”: Please change the order based on the appearance in the manuscript. It should be as “XRD, SEM, N2GA and NMR”.

Reply:

Thanks for your valuable suggestions. We have modified the order according to the appearance of the manuscript (Please see Lines #91).

Q5: Under 2.21: Please use two paragraphs.: One for XRD explanations and another one for SEM explanations.

Reply:

Thanks for your valuable suggestions. We have used two paragraphs in the manuscript to explain XRD and SEM respectively (Please see Lines #95-103).

Q6: Line 132: What is the pressure unit?

Reply:

Thanks for your valuable suggestions. The pressure in the manuscript is the relative pressure (P/P_0) as a dimensionless unit (Please see Lines #123).

Q7: Line 138: BET, STP, BJH:Please use the full names and give the abbreviations in parenthesis.

Reply:

Thanks for your valuable suggestions. We have given the full name in the manuscript and used abbreviations in parentheses (Please see Lines #129-131).

Q8: Lines 81, 119, 132, 137, 140, 190, 215, 229, 252: “We”: Please do not use active voice in the manuscript, you should use passive voice.

Reply:

Thanks for your valuable suggestions. We have revised all the active voice in the manuscript to passive voice (Please see Lines #71, 110, 123, 128, 132-133, 185-186, 209-210, 226-227, 251-253).

Q9: Table 3: Please do not use Chinese in the manuscript, change it to English.

Reply:

Sorry, this is our mistake. We have changed the Chinese in Table 3 to English (Please see Table 3).

Q10: Line 189: Explain the first sentence: What is Washburn equation?

Reply:

This is a good suggestion. We have put the Washburn equation in the manuscript (Please see Lines #182-184).

Q10: Line 203: What does very good mean quantitatively for peaks of T₂ spectrum and connectivity?

Reply:

Thanks for your valuable suggestions. We apologize for our improper expression. In fact, the T₂ spectrum is directly related, and the wider the T₂ spectrum, the better the connectivity of the surface pores. We have made changes in the manuscript (Please see Lines #195-198).

Q11: Figure 7: Does magenta curve belong to Sample 4? It has been written S7. Please fix this issue.

Reply:

Thanks for your valuable suggestions. We are very sorry for our incorrect representation of the sample name in Figure 7. We have changed S7 in Figure 7 to S4 (Please see Figure 7).

Q12: While giving the average values in the text, include standard deviation in the parentheses (i.e., Average (±SD) Unit).

Reply:

Thanks for your valuable suggestions. We have added the standard deviation after the average (Please see Lines #138, 140, 266-271).

Q13: Please fix, misspelled words in the text.

Reply:

Thanks for your valuable suggestions. Through re-reading the full paper carefully, all of the grammatical and spelling errors have been revised in the revised manuscript (Please see Lines #48, 265, etc.).

Q14: Line 373 Please explain your last conclusion paragraph sentences. “The interlayer homogeneity of this kind of uranium-bearing sandstone is good, and the difference in porosity and

permeability is not large. The same well spacing can be used to improve the production efficiency.”

Reply:

Thanks for your valuable suggestions. the better homogeneity of this sandstone will make the seepage traces in all directions more uniform, so the same well spacing can be used to improve production efficiency (Please see Lines #375-378).

Finally, we appreciate your constructive suggestions and pertinent comments, which are of importance to improve the quality of our manuscript. Thank you very much!

Reviewer: 2

Comments:

Q1: The first paragraph of Introduction, too long, cut down.

Reply:

We appreciate you very much for the valuable comments. We have deleted part of the content of the first paragraph of the introduction (Please see lines #28-45).

Q2: Paragraph 2, about the methods in pore structure characterization, see: Lai J., Wang G., Wang Z., Chen J., Pang X., Wang S., Zhou Z., He Z., Qin Z., Fan X. 2018. A review on pore structure characterization in tight sandstones. Earth-Science Reviews, 177, 436–457.

Reply:

We appreciate you very much for the valuable comments. We have carefully studied the paper you provided, which is an important supplement to the characterization method of the pore structure in our manuscript (Please see lines #50-51).

Q3: Section 2.1 Geological settings should be a new section.

Reply:

We appreciate you very much for the valuable comments. We have changed section 2.1 Geology to a new chapter (Please see line #81).

Q4: Section 3. Had better divided results from Discussion.

Reply:

Thanks for your valuable comments. We have separated the results of the discussion(Please see line #152, 185, 191, 222, 241, 265, 283, 294, 345).

Q5: Section 3.1 I think you firstly should use thin sections.

Reply:

Thanks for your valuable comments.We have made the sample into thin slices before using the SEM (Please see line #137).

Q6:There are many long paragraphs.

Reply:

Thanks for your valuable comments. We have changed the long paragraph to the short paragraph (Please see line #152, 185, 191, 222, 241, 265, 283, 294, 345).

Q7: NMR capture continuous whole pore body distribution (see: Lai J., Wang S., Zhang C., Wang G., Song Q., Chen X., Yang K., Yuan C. 2020. Spectrum of pore types and networks in the deep Cambrian to Lower Ordovician dolostones in Tarim Basin, China. Marine and Petroleum Geology, 112, 104081.), but N₂ga is not, how to combine.

Reply:

Thanks for your valuable comments. We have conducted a careful study of the paper you provided. This is an important supplement to our manuscript. The NMR test has a wide range of pore size measurement, but the accuracy of some pores is not high, and the nitrogen adsorption experiment can just make up for it (Please see line #210-211).

Q8: Too many equations in the paper, and references may need if they are not developed by the authors themselves.

Reply:

Thanks for your valuable comments. We have added the corresponding references to the equation (Please see line #309, 316).

Special thanks to you for your good comments.